

# Consequences and mitigation of saltwater intrusion induced by short-circuiting during aquifer storage and recovery (ASR) in a coastal subsurface

Koen Gerardus Zuurbier[1,2], Pieter Jan Stuyfzand[1,2]

[1] KWR Watercycle Research Institute, Groningenhaven 7, 3433 PE, The Netherlands
[2] Technical University Delft, Faculty of Civil Engineering, P.O. Box 5048, 2600 GA Delft, The Netherlands

*Correspondence to*: Koen Zuurbier (koen.zuurbier@kwrwater.nl)

**Abstract.** Coastal aquifers and the deeper subsurface are increasingly exploited. The accompanying perforation of the subsurface for those purposes has increased the risk of short-circuiting of originally separated aquifers. This study shows how this short-circuiting negatively impacts the freshwater recovery efficiency (RE) during aquifer storage and recovery (ASR) in coastal aquifers. ASR was applied in a shallow saltwater aquifer overlying a deeper saltwater aquifer, which was

targeted for seasonal aquifer thermal energy storage (ATES). Although both aquifers were considered properly separated, intrusion of deeper saltwater into the shallower aquifer quickly terminated the freshwater recovery. The presumable pathway was a nearby ATES borehole. This finding was supported by field measurements, hydrochemical analyses, and SEAWAT transport modelling. The potentially rapid short-circuiting during storage and recovery can reduce the RE of ASR to null. When limited mixing with ambient groundwater is allowed, a linear RE decrease by short-circuiting with increasing distance

from the ASR well within the radius of the injected ASR-bubble was observed. Interception of deep short-circuiting water can mitigate the observed RE decrease, although complete compensation of the RE decrease will generally be unattainable. Brackish water upconing from the underlying aquitard towards the shallow recovery wells of the MPPW-ASR system was observed. This 'leakage' may lead to a lower recovery efficiency than based on current ASR performance estimations.

KEYWORDS: aquifer storage and recovery, ASR, saltwater intrusion, coastal aquifers, short-circuiting, hydraulic connections, numerical modelling

## 1 Introduction

Aquifers are increasingly being used for stormwater infiltration (Ferguson, 1990), brine disposal (Stuyfzand and Raat, 2010; Tsang et al., 2008), and storage of freshwater (aquifer storage and recovery or ASR; Pyne, 2005), heat (aquifer thermal

energy storage or ATES; Bonte et al., 2011a), and $CO_2$ (Steeneveldt et al., 2006). Additionally, they are perforated for exploitation of deep fossil and geothermal energy and traditionally used for abstraction of drinking and irrigation water. The increased use of the subsurface can lead to interference among aquifer storage systems (e.g., Bakr et al., 2013) or affect the





groundwater quality (Bonte et al., 2013a; Bonte et al., 2011a; Bonte et al., 2011b; Bonte et al., 2013b; Zuurbier et al., 2013b). These consequences form relevant fields of current and future research.

The perforation of aquifers and aquitards accompanying the subsurface activities imposes an additional risk by the potential creation of hydraulic connections ('conduits') between originally separated aquifers or aquifers and surface waters. This risk

is plausible, as estimations indicate that about two-thirds of the wells may be improperly sealed (Morris et al., 2003), although the attention for this potential risk is limited (Chesnaux, 2012). Additionally, many of the new concepts to use the subsurface (e.g., ATES, ASR, brine disposal) require injection via wells, which may cause soil fractures, even when the annulus is initially properly sealed, by exceedance of the maximum-permissible injection pressure (Hubber and Willis, 1972; Olsthoorn, 1982). The soil fractures are undesirable for most groundwater wells in the relatively shallow subsurface, since

they create new connections between originally separated aquifers.

The resulting short-circuiting or leakage process has been studied at laboratory (Chesnaux and Chapuis, 2007) and field scale (Jiménez-Martínez et al., 2011; Richard et al., 2014), and for deep geological $CO_2$ storage (Gasda et al., 2008). Santi et al. (2006) evaluated tools to investigate cross-contamination of aquifers. Chesnaux et al. (2012) used numerical simulations of theoretical cases to demonstrate the consequences for pumping test and hydrochemistry of hydraulic connections between

granular and fractured-rock aquifers, which clearly demonstrated the significant hydrochemical cross-contamination when short-circuiting aquifers have a distinct chemical composition. The impact of short-circuiting on ASR are however not evaluated to date. However, reliably confined aquifers are vital to successfully store energy (Bonte et al., 2011a) and freshwater (Zuurbier et al., 2013a; Zuurbier et al., 2015; Zuurbier et al., 2014) to bridge periods of surplus and demand. And although the risks of short-circuiting by perturbation are acknowledged by scientists, it seems that the practical and

regulatory communities are less aware (Chesnaux, 2012). This is underlined by the fact that certification for mechanical drilling (applied since the Industrial Revolution) in The Netherlands was not obliged before 2011 (Stichting Infrastructuur Kwaliteitsborging Bodembeheer, 2013a), while for the subsurface design and operation of ATES systems (>1500 systems since the nineties (Bonte et al., 2011a; CBS, 2013)) certification was obliged only since early 2014 (Stichting Infrastructuur Kwaliteitsborging Bodembeheer, 2013b).

The lack of proper design and regulation of subsurface activities using wells can be partly caused by the lack of clear field examples of how well-intentioned use of the subsurface for sustainability purposes can fail thanks to earlier activities underground. This lack can be caused by the fact that short-circuiting may not be easy to observe (Santi et al., 2006), or because failing or disappointing projects often do not make it to public or scientific reports. Therefore, we present in this study how short-circuiting via a deeper borehole led to failure of freshwater recovery during ASR in a coastal aquifer. The

objective of this paper is to demonstrate and characterize the potential consequences of perturbations for coastal aquifer storage and recovery (ASR) systems. Additionally, the use of deep interception of saltwater to improve shallow recovery of freshwater upon ASR was assessed. The Westland ASR site in the coastal area of The Netherlands served as demonstration and reference case.



## 2    Methods

### 2.1    Set-up Westland ASR system

The Westland ASR system is installed to inject the rainwater surplus of 270,0000 m$^2$ of greenhouse roof in a local shallow aquifer (23 to 37 m-below sea level (m-BSL), surface level = 0.5 m-above sea level (m-ASL)) with negligible lateral

displacement (Zuurbier et al., 2013a) for recovery in times of demand. For this purpose, two multiple partially penetrating wells (MPPW) were installed (Figure 1), such that water can be injected preferably at the aquifer base, and recovered at the aquifer top in order to increase the recovery (Zuurbier et al., 2014). All ASR (AW1 and AW2, installed in 2012) and ATES (K3, installed in 2006 and replaced close by in 2008) wells were installed using reverse-circulation rotary drilling, while the monitoring wells (MW1-5, Figure 2) were installed using bailer drillings. Bentonite clay was applied to seal the ASR wells

(type: Micolite300) and ATES well K3 (Micolite000 and Micolite300). The ASR wells used a 3.2 m high standpipe to provide injection pressure, whereas the ATES well used a pump to meet the designed injection rate of 75 m$^3$/h. The maximum Cl concentration in the recovered water accepted at the site is 50 mg/l, which meant that the water should be recovered practically unmixed.

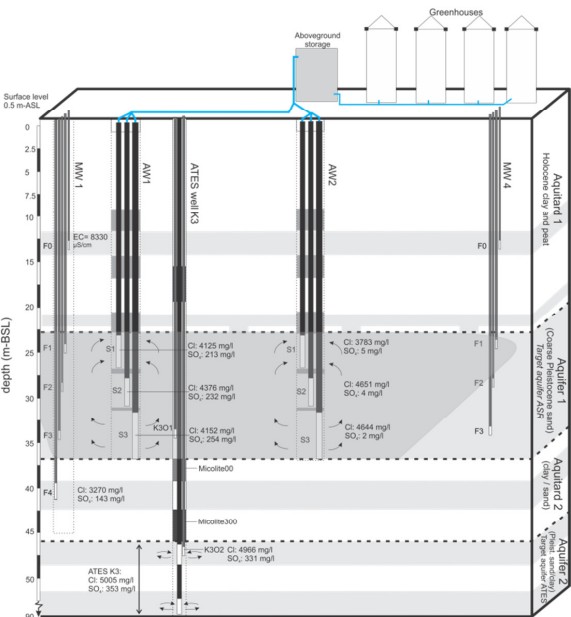

**Figure 1: Cross-section of the Westland ASR site to schematize the geology, ASR wells, ATES well, and the typical hydrochemical composition of the native groundwater. Horizontal distances not to scale.**





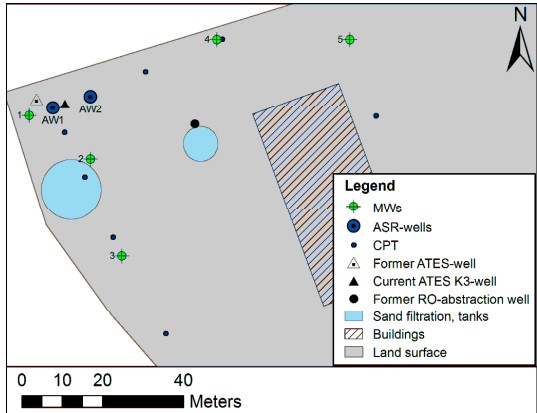

Figure 2: Locations of ASR (AW), ATES, and monitoring wells (MW).

## 2.2 Monitoring during Westland ASR cycle testing

All ASR and monitoring well screens were sampled prior to ASR operation (November and December, 2012). MW1 and 2
were sampled with a high frequency during the first breakthrough of the injection water at MW1 (December 2012, January
2013), while all wells were sampled on a monthly basis. Three times the volume of the well casing was removed prior to
sampling. The injection water was sampled regularly during injection phases. All samples were analyzed in the field in a
flow-through cell for EC (GMH 3410, Greisinger, Germany), pH and temperature (Hanna 9126, Hanna Instruments, USA),
and dissolved oxygen (Odeon Optod, Neotek-Ponsel, France). Samples for alkalinity determination within one day after
sampling on the Titralab 840 (Radiometer Analytical, France) were stored in a 250 ml container. Samples for further
hydrochemical analysis were passed over a 0.45 μm cellulose acetate membrane (Whatman FP-30, UK) in the field and
stored in two 10-ml plastic vials, of which one was acidified with 100 μl 65% $HNO_3$ (Suprapur, Merck International) for
analysis of cations (Na, K, Ca, Mg, Mn, Fe, S, Si, P, and trace elements) using ICP-OES (Varian 730-ES ICP OES, Agilent
Technologies, U.S.A.). The second 10 ml vial was used for analysis of F, Cl, $NO_2$, Br, $NO_3$, $PO_4$, and $SO_4$ using the Dionex
DX-120 IC (Thermo Fischer Scientific Inc., USA), and $NH_4$ using the LabMedics Aquakem 250 (Stockport, UK). All
samples were cooled to 4 °C and stored dark immediately after sampling.

CTD-divers (Schlumberger Water Services, Delft, The Netherlands) were used for electronic recording of conductivity,
temperature, and pressure in the target aquifer at MW1 and MW2. Calibrated, electronic water meters were coupled to the
programmable logic controller (PLC) of the ASR system to record the operation per well screen.

## 2.3 Set-up Westland ASR groundwater transport model

In SEAWAT Version 4 (Langevin et al., 2007) was used to with PMWIN 8 (Chiang, 2012)simulate the ASR operation. A
half-domain was modelled to reduce computer runtimes. Cells of 1x1 m were designated to an area of 20x20 m around the





ASR wells. The cell size increased to 2.5x2.5 m (30x40 m around te well) and was then gradually increased to a maximal cell size of 200x200 m at 500 m from the ASR wells. The pumping rate of each well screen was distributed over the models cells with the well package based on the transmissivity (thickness x hydraulic conductivity) of each cell. The third-order total-variation-diminishing (TVD) scheme (Leonard, 1988) was used to model advection.

Equal constant heads were imposed at two side boundaries of the aquifers, the top of the model (controlled by drainage) and at the base of the model. No-flow boundaries were given to the other two side boundaries of the model. Initial Cl-concentrations were based on the results of the reference groundwater sampling at MW1. SO₄ concentrations in Aquifer 1 were based on MW2, since these concentration were more representative for the field site. For Aquifer 2, the concentrations found at ATES well K3 (bulk) and the observation well K3.1 were used (see Figure 1). The density of the groundwater was

based on the Cl concentration using:

$$\rho_w = 1000 + 0.00134 \times Cl(mg \, / \, l)$$

Density and viscosity were not corrected for temperature, as all temperatures (background groundwater, injected ASR water, and injected ATES water) were in the range of 8 to 12 °C and should not significantly impact the flow pattern (Ma and

Zheng, 2010). A longitudinal dispersivity of 0.1 m was derived from the freshwater breakthrough at MW1 and was applied to the whole model domain. Constant heads were based on the local drainage level (top model layer) and the observed heads in the aquifer. The regional hydraulic gradient was derived from regional groundwater heads (TNO, 1995)

The recorded pumping rates of the ASR wells and the ATES K3 well during two ASR cycles were incorporated in the SEAWAT model. The ASR operation was modelled with a properly sealed and an unsealed ATES borehole. In the latter

case, a hydraulic conductivity (K) of 1000 m/d was given to the cells (1.0 x 1.0m) in Aquifer 1, Aquitard 2, and Aquifer 2 at the location of the ATES pumping well to force a significant borehole leakage. This K was considered realistic since apart from filter sand around the well screen, the borehole was backfilled with gravel with a grain size of 2 to 5 mm. In later scenarios, the ATES well was moved towards the fringe of the ASR well stepwise (10 m further away from AW1 in each scenario), after which Cycle 2 was simulated again. This was to examine the impact of borehole leakages at various

distances from the ASR-wells.




**Table 1: Hydrogeological properties of the geological layers in the Westland SEAWAT model**

| Geological Layer | Model layers | Base (m-BSL) | $K_h$ (m/d) | VANI $(K_h/K_v)$ | Ss $(m^{-1})$ | n (-) | Initial C (mg/l Cl) | Initial C (mg/l $SO_4$) |
|---|---|---|---|---|---|---|---|---|
| Aquitard 1 | 6 | 22.3 | 0.2 - 1 | 100 | $10^{-4}$ | 0.2 | 2000-3000 | 4 |
| Aquifer 1 | 12 3 | 33.7 36.4 | 35 100 | 1 | $10^{-7}$ | 0.3 | 4000-4800 | 4 |
| Aquitard 2 (clay-sand) | 8 | 47.5 | 0.05-10 | 1-10 | $10^{-4}$ | 0.2-0.3 | 3200 | 160 |
| Aquifer 2 | 6 | 96 | 12 | 1 | $10^{-6}$ | 0.3 | 4100-7900 | 331-375 |

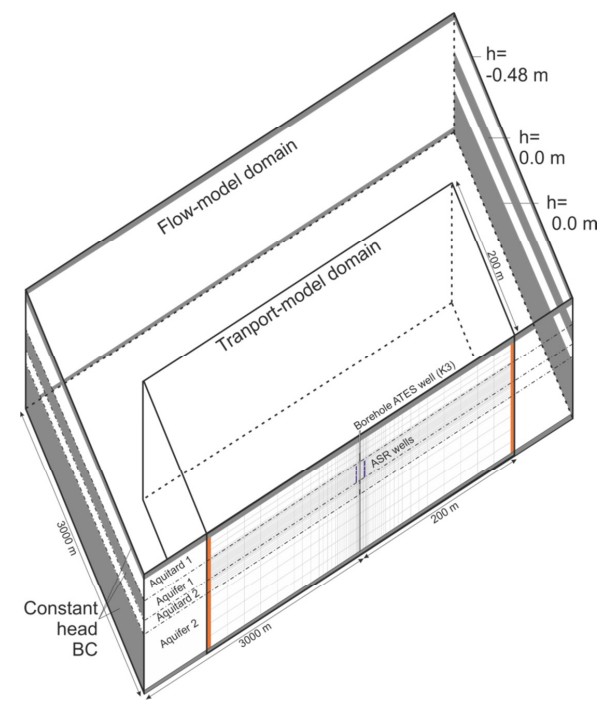

**Figure 3: Set-up of the Westland ASR groundwater transport model (half-domain).**

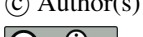



### 2.4    The maximal recovery efficiency with and without leakage at the Westland ASR site.

The collected data on the aquifer characteristics in the SEAWAT groundwater model were used to analyze the future performance of the MPPW-ASR system for the current (with leakage) and a 'normal field site' (without leakage from deeper aquifers via a perturbation, or after sealing of the perturbation). The SEAWAT model was used to simulate three consecutive

ASR-cycles with the representative operational characteristics from Table 2 for the Westland site (Zuurbier et al., 2012). Once the recovered Cl concentration exceeded 50 mg/l, the model was stopped, and the length of the stress period with recovery was adjusted, such that no water with Cl>50 mg/l was recovered. Subsequently the model was run again after adding another cycle.

**Table 2: Set-up of the modelled, representative ASR-cycle for the Westland subsurface without short-circuiting of deeper saltwater.**

| Stage | Duration | Pumping rate |
|---|---|---|
| Infiltration | 120 days | 60,000 / 120 = 500 m$^3$/d |
| Storage | 30 days | 0 m$^3$/d |
| Recovery | 120 days | -60,000 / 120 = -500 m$^3$/d |
| Idle | 65 days* | 0 m$^3$/d |

\* Longer when early salinization occurred during recovery.

## 3    Results

### 3.1    Detailed hydrogeological characterization based on local drillings

The target aquifer for ASR (Aquifer 1) was found to be 14 m thick and consists of coarse fluvial sands (average grain size: 400 µm) with a hydraulic conductivity (K) derived from head responses at the monitoring wells upon pumping of 30 – 100 m/d. Aquifer 2 (target aquifer for ATES) has a thickness of more than 40 m, but is separated in two parts at the ATES well

K3 by a 20 m thick layer clayey sand and clay. A blind section was installed in this interval, and the borehole was backfilled with coarse gravel in this section. The K-value of the fine sands in Aquifer 2 derived from a close by pumping test is 10 to 12 m/d and is in line with the estimated K-value from grain size distribution (Mos Grondmechanica, 2006). The effective screen length of K3 in this aquifer is only 8 (upper section: 53-61 m-BSL) and 5 m (lower section: 80-85 m-BSL).

The groundwater is typically saline, with observed Cl concentrations ranging 3,793 to 4,651 mg/l in Aquifer 1 and

approximately 5,000 mg/l in Aquifer 2 (see also Figure 1). A sand layer in Aquitard 2 contains remnant fresher water (Cl = 3,270 mg/l). SO$_4$ is a useful tracer to identify the saltwater from Aquifer 1 and 2: it is virtually absent in Aquifer 1




(presumably younger groundwater, infiltrated when the Holocene cover was already thick), whereas it is high in Aquifer 2 (older water, infiltrated through a thinner clay cover which limited $SO_4$-reduction, see Stuyfzand (1993) for more details): 300 to 400 mg/l $SO_4$.

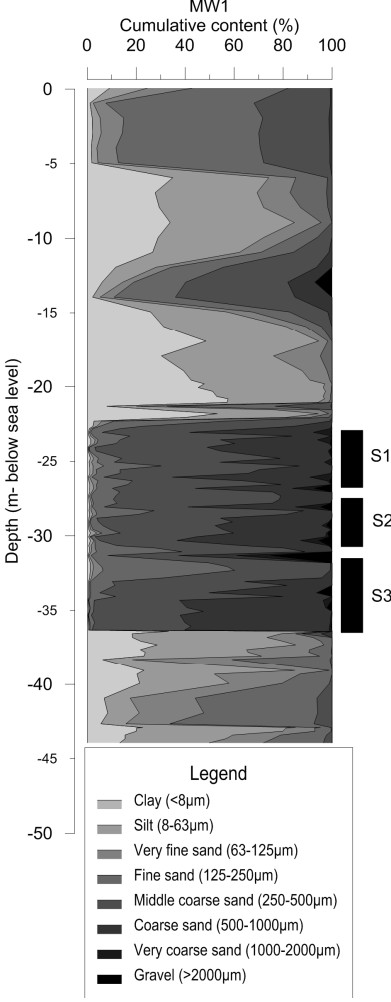

5 Figure 4: Cumulative grain size contents observed at MW1 (at 5 m from ASR well 1) in this study. S1-S3 mark the depth intervals of the ASR well screens.


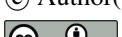

### 3.2    Cycle 1 (2012/2013): first identification of borehole leakage

The first ASR-cycle started in December 2012. The first recovery started halfway January 2013. Despite the abstraction with only the shallow wells of the MPPW, a rapid and severe salinization was found within the first days of recovery, after injecting freshwater for about 1 month (Figure 5). Remarkably, the salinization at ASR well 1 (AW) preceded salinization

5      of the monitoring wells situated further from the ASR wells (MW1, MW2). High $SO_4$ concentrations (up to >50 mg/l) were found in the recovered water, which could not be explained by the $SO_4$-concentration attained by pyrite oxidation by oxygen and nitrate present in the injection water (Zuurbier et al., 2016), which would result in $SO_4$ concentrations of less than 15 mg/l.

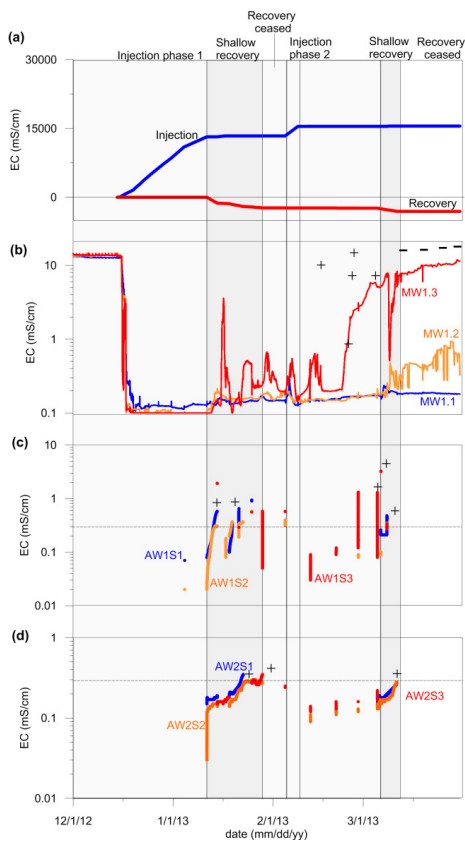

10    **Figure 5: Pumping of the ASR system during cycle 1 (2012/2013), EC observations at MW1 (5 m from AW1), and the EC in the recovered water at AW1 and AW2. MW= monitoring well, AW= ASR well.**




The SEAWAT model underlined that tilting of the freshwater-saltwater interfaces at the fringe of the ASR bubble did not cause the early salinization observed, as this would have led to a much later salinization (Figure 6) without enrichment of $SO_4$ (other than be pyrite oxidation), even if the recovery period was extended (results not shown). When the leaky borehole was incorporated in the model (by assigning K=1000 in a 1 x 1 m column at the location of the current ATES well), it was

5  able to introduce the early recovery of deep ($SO_4$-rich) water (Figure 7). Other scenarios that were tested, but unable to improve the simulation of the observed $SO_4$-trends were: leakage from via the former ATES K3 well further from the ASR wells (arrival of $SO_4$ too late), a high-K borehole (2000 m/d; arrival too early, flux too high), a low-K borehole (500 m/d; arrival too late, flux too low), a vertical anisotropy in the aquifers ($K_h/K_z = 2$; arrival too early, flux too high), and omission of the deep cold water abstraction from Aquifer 2 via the ATES well in Aquifer 2 ($SO_4$-flux too high).

10  The hydrochemical observations and model outcomes of Cycle 1 indicated that the source of the early salinization was the intrusion of saltwater from Aquifer 2. Considering the lithology, thickness, and continuity of Aquitard 2 (confirmed by grain size analyses and cone penetrating tests on the site), leakage via natural pathways through this separating layer was unlikely. According to the rate and sequence of salinization, the leakage could well be situated at the ATES K3 well close to AW1.

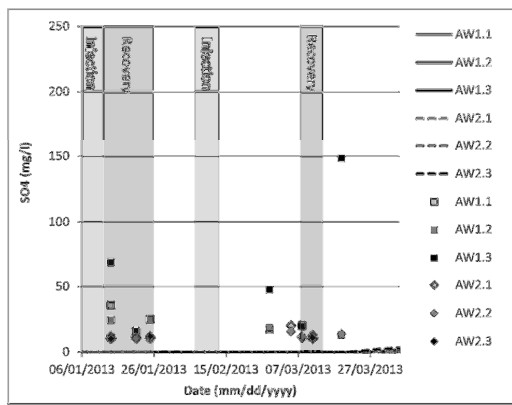

**Figure 6: Modelled (solid lines) and observed (data points) $SO_4$ concentrations without borehole leakage. High concentrations indicate admixing of deeper saltwater. Observed $SO_4$ concentrations by far exceed the modelled concentrations.**





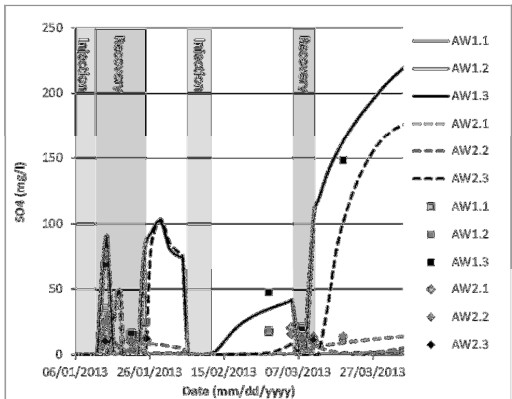

**Figure 7: Modelled (solid lines) and observed (data points) SO$_4$ concentrations. Borehole leakage at the location of the current ATES K3 well via a 1x1 m borehole with K=1000 m/d. High concentrations indicate admixing of deeper saltwater. Observed SO$_4$ concentrations become in line with the modelled concentrations.**

### 3.3    Cycle 2 (2013/2014): improving the ASR operation

Cycle 2 started with the injection of 66,178 m$^3$ of rainwater using both ASR wells between September 2013 and March 2014, which was followed by recovery solely at the downstream AW2 (start: March 5, 2014). A rapid salinization by SO$_4$-rich saltwater was again observed (Figure 8) and the recovery was terminated after 26 days (March 21, 2014) after

recovering no more than 2,500 m$^3$. During this cycle, a monitoring well present in the gravel pack of the ATES K3 well (coded K3O1; a 1m-well screen at 33 m-BSL) was also sampled and equipped with a CTD-diver and continuously pumped with a rate of 1 m$^3$/h, unraveling high ECs and presence of SO$_4$-rich saltwater from the deeper aquifer in the centre of the injected freshwater body (Figure 8). This presence of intruding deep saltwater was also found at MW1S3 (5m from the ASR wells) as a consequence of displacement while re-injecting part of the abstracted freshwater from the shallow AW2S1 wells

screen at the deeper AW2S3 well screen and density-driven flow (spreading over de base of the aquifer). The observed Cl concentration (268 mg/l) on April 2, 2014 at MW1S4 (situated in Aquitard 2 at 5 m from AW1) was significantly lower than at MW1S3 (2,528 mg/l) and K3O1 (3,341 mg/l), indicating that salinization of the shallow target aquifer (Aquifer 1) preceded salinization of Aquitard 2.

In order to re-enable recovery of freshwater, the deepest wells of the MPPWs (AW1S3 and AW2S3) were transformed to

interception wells or 'Freshkeepers' (Stuyfzand and Raat, 2010; Van Ginkel et al., 2014), abstracting the intruding saltwater and injecting this in a deep injection well in Aquifer 2 at 200 m distance from the ASR-site. This way, an acceptable water quality (Cl <50 mg/l) could be recovered at AW2S1 and AW1S2 again (from April 15, onwards). As a consequence, the deeper segments of the target aquifer (S3 levels, Figure 8bcd) first freshened, followed by again salinization as recovery





proceeded. Saline water was continuously observed at K3O1, indicating that leakage via the K3 borehole continued. After recovery of in total 12,324 m$^3$ of practically unmixed rainwater (18.6% of the injected water), the recovery had to be ceased due to the increased salinity. During this last salinization, the water at the deep (S3-)levels of the target aquifer at AW1, MW1, and MW2 showed low $SO_4$-concentrations, indicating salinization by saltwater from Aquifer 1 instead of deep

5    saltwater from Aquifer 2. High $SO_4$-concentrations (>100 mg/l) were only found close to the current K3 ATES well (the presumable conduit) in this phase (AW1 and K3O1).

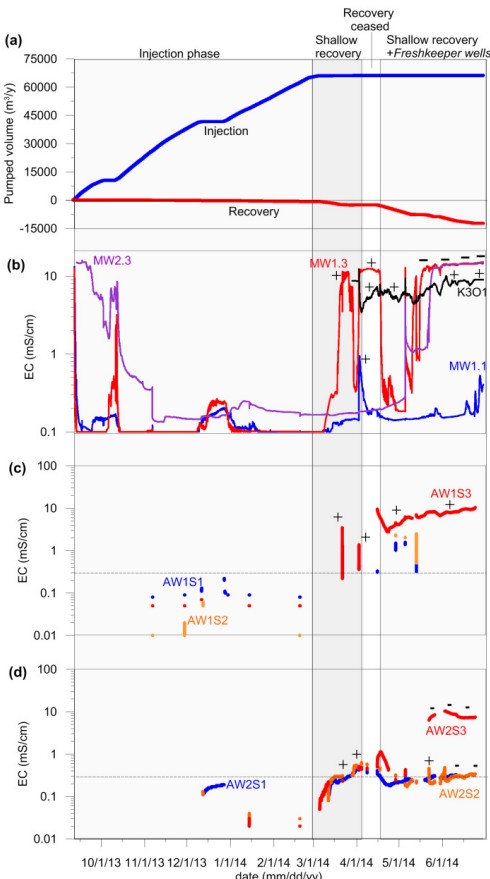

**Figure 8: Pumping of the ASR system during cycle 2 (2013/2014), EC observations at MW1 (5 m from AW1), and the EC in the**
10    **recovered water at AW1 and AW2. AW2.1 and AW2.3 were used for freshwater recovery (12,324 m$^3$). Presence of increased $SO_4$-**
    **concentrations (deep saltwater) are marked by '+', while its absence is marked by '-' (indicating shallow saltwater).**




The SEAWAT model with leakage via the borehole of the current ATES well K3 was able to reasonably simulate the water quality trends regarding $SO_4$ and Cl in Cycle 2 (Figure 9 and Figure 10). Remaining deviations in observed concentrations were contributed to uncertainties in the model input, mainly aquifer heterogeneity, potential stratification of the groundwater quality in Aquifer 2, and disturbing abstractions and injections in the surroundings, mainly by nearby ATES and brackish

5    water reverse osmosis systems, the latter abstracting from Aquifer 1 and injecting in Aquifer 2.

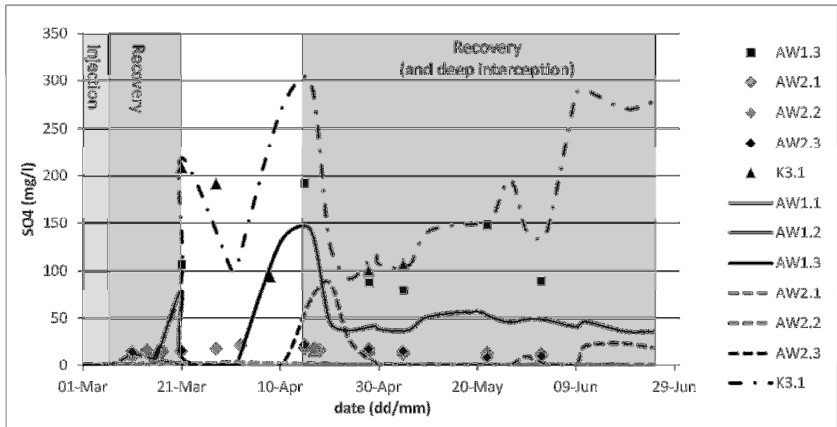

**Figure 9: Modelled and observed $SO_4$-concentrations at the most relevant well screens.**

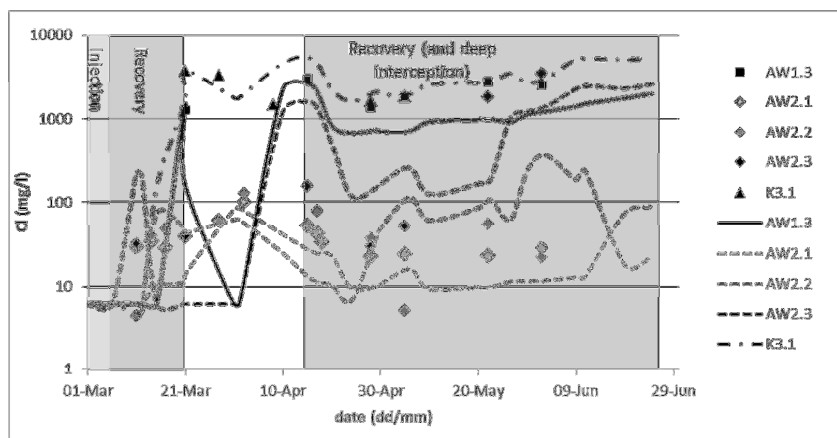

**Figure 10: Modelled and observed Cl-concentrations at the most relevant well screens.**



Modelling of Cycle 2 demonstrated that salinization during recovery was independent of the injected freshwater volume. Salinization occurred after recovery with the same rate as in Cycle 1, despite a four times larger injection volume. Analysis of the modelled concentration distribution and pressure heads showed that injected freshwater could not reach deep into the deeper saline aquifers since the freshwater head in the leaky ATES borehole during injection was more or less equal to the

freshwater head in the deeper saltwater aquifer. In other words: little freshwater was pushed through the conduit into the deeper aquifer. Further on, the freshwater that did reach the deeper aquifer got rapidly displaced laterally as a result of buoyancy effects (Figure 11).

A significant head difference ($\Delta h$(fresh)= 0.3 m to 0.65 m) was observed in the model during recovery. In combination with the high permeability of the ATES borehole, this resulted in a significant intrusion of deeper ($SO_4$-rich) saltwater. Even

during storage phases, a freshwater head difference ($\Delta h$(fresh)= 0.15 m) was observed as a consequence of replacement of saltwater by freshwater in the target aquifer, causing intrusion of deep saltwater, yet with a lower rate than during recovery.

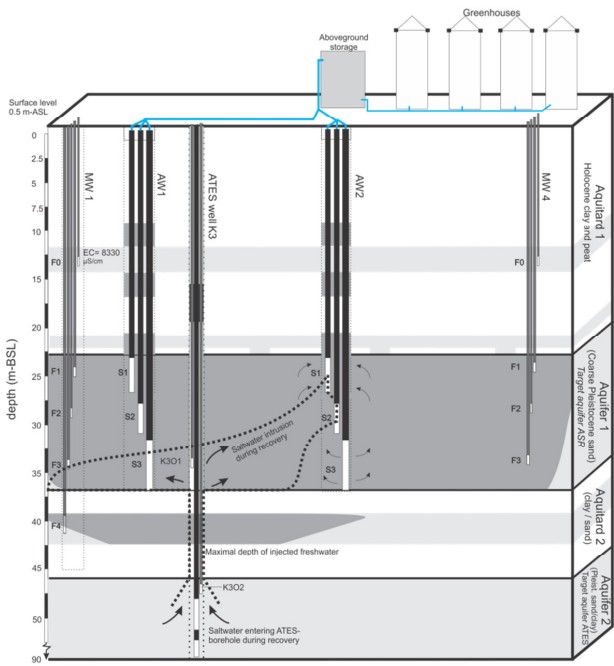

**Figure 11: Deep saltwater intrusion via the current ATES K3 borehole during shallow recovery of injected freshwater at the**
**Westland ASR site at the start of Cycle 2.**



### 3.4 Analysis of the leakage flux via the borehole

An analytical solution was presented by Maas (2011) to calculate the vertical leakage via a gravel or sand pack. In this solution, it is presumed that an aquitard was pierced during drilling and the annulus was filled up with sand or gravel without installing a clay seal. The leakage is then calculated as function of the different hydraulic conductivities, pressure difference,

and the radius of the borehole and well screen:

$$Q_{VGP} = \frac{\Delta h_{GP}}{W}$$

where: $Q_{VGP}$ = vertical leakage via gravel pack (m$^3$/d), $\Delta h_{GP}$ = hydraulic head difference between 2 sections of the gravel pack, one being the inflow and the other the outflow section (m), and $W$ = leakage resistance (d/m$^2$) and is calculated as:

$$W \approx \frac{(0.005(\ln(\alpha))^2 - 0.058\ln(\alpha) + 0.19)}{(r_1\sqrt{K_{HIN}K_{VIN}})}$$

And $a$ as:

$$\alpha = \frac{K_{VGP}(r_1^2 - r_0^2)}{2K_{VIN}r_1^2} /$$

where: $r_0$ = radius of well screen [m]; $r_1$ = radius of borehole [m]; $K_{VGP}$ = vertical hydraulic conductivity of gravel pack [m/d]; $K_{VIN}$ = vertical hydraulic conductivity of inflow aquifer layer [m/d]; $K_{HIN}$ = horizontal hydraulic conductivity of the inflow aquifer layer [m/d].

Calculating the leakage flux using the $\Delta h_{GP}$ from the SEAWAT model underlines that the pressure differences induced by density differences and enhanced during abstraction for freshwater recovery in combination with an unsealed borehole leads to a saltwater intrusion ($Q_{VGP}$) of around 50 to 200 m$^3$/d, which is in line with the observed leakage flux in the SEAWAT model.





Table 3: Calculated leakage flux $Q_{VGP}$ via the (unsealed) borehole based on Maas (2011) for different net recovery rates ($Q_{recovery, net}$).

|  |  | Storage (no recovery) | Low recovery rate | High recovery rate |
|---|---|---|---|---|
| $Q_{recovery, net}$ | $(m^3/d)$ | 0 | 77 | 371 |
| $\Delta h_{GP}$ | $(m)$ | 0.15 | 0.30 | 0.66 |
| $Q_{VGP}$ | $(m^3/d)$ | 49 | 99 | 215 |
|  |  |  |  |  |
| W | $(m^2/d)$ | 0.0031 | 0.0031 | 0.0031 |
| $\alpha$ |  | 4.7 | 4.7 | 4.7 |
| $r_0$ | $(m)$ | 0.1 | 0.1 | 0.1 |
| $r_1$ | $(m)$ | 0.4 | 0.4 | 0.4 |
| $K_{HIN}$ | $(m/d)$ | 100 | 100 | 100 |
| $K_{VIN}$ | $(m/d)$ | 100 | 100 | 100 |
| $K_{VGP}$ | $(m/d)$ | 1000 | 1000 | 1000 |

### 3.6 The maximal recovery efficiency with and without leakage at the Westland ASR site.

The SEAWAT model was used to evaluate the ASR performance at the Westland field site with three different ASR strategies (Table 4), with and without the saltwater leakage. During the 120 days of recovery it was aimed to recover as much of freshwater (marked by Cl <50 mg/l) as possible. Equal abstraction rates were maintained for both ASR wells (AW1 and AW2) in the scenario's without leakage, while only AW2 was used for recovery in the scenario's with leakage.

Recovery with conventional, fully penetrating ASR wells will be limited to around 30% of the injected freshwater in a case without the saltwater leakage. For the case with leakage, freshwater recovery will be impeded by the short-circuiting during the storage phase: the wells will produce brackish water already at the start of the recovery phase. The use of a MPPW for deep injection and shallow recovery has a limited positive effect due to the limited thickness of the aquifer: one-third of the injected water can be recovered in a case without leakage. The improvement of RE by introduction of the MPPW is limited in comparison with the conventional ASR well since some saltwater from Aquitard 2 was found to move up to the shallower recovery wells of the MPPW-system ('upconing') rapidly after the start of recovery. The slight increase in Cl concentrations caused by this process is sufficient to terminate the recovery due to exceedence of the salinity limit. Before the fringe of the freshwater bubble reached the recovery wells, recovery was already terminated. In the case of saltwater leakage, salinization occurred within 2 days, limiting the RE to only 1%.



**Table 4: Modelled recovery efficiencies at the Westland ASR site without short-circuiting using different pumping strategies. The relative pumping rate per MPPW well screen is given for each particular screen.**

| Strategy | Distribution pumping rate | RE (short-circuiting / no short-circuiting) | Intercepted brackish-saline water (via deep (S3-)wells) |
|---|---|---|---|
| Conventional ASR-well | In: 100% via one fully penetrating well | Year 1: 0/15% | |
| | Out: 100% via one fully penetrating well | Year 2: 0/25% | |
| | | Year 3: 0/30% | |
| | | Year 4: 0/32% | |
| Deep injection, shallow recovery (MPPW-ASR) | In: 10/20/70% (Year 1) | Year 1: 1/19% | |
| | In : 0/20/80% (Year 2-3) | Year 2: 1/ 29% | |
| | Abstract: 60/40/0% (Year 1-3) | Year 3: 1/32% | |
| | | Year 4: 1/33% | |
| MPPW-ASR + 'Freshkeeper' | In: 10/20/70% (Year 1) | Year 1: 29/40% | Year 1: 32,700/ 18,500 m$^3$ |
| | In : 0/20/80% (Year 2) | Year 2: 32/46% | Year 2: 33,000 / 20,500 m$^3$ |
| | Abstract: Decreasing from 60/40/0% to | Year 3: 33/47% | Year 3: 31,900 / 21,500 m$^3$ |
| | 60/0/0% (Year 1-3) | Year 4: 33/48% | Year 4: 31,500 / 19,300 m$^3$ |
| | Intercept Freshkeeper: increasing from 100 to 500 m$^3$/d | | |




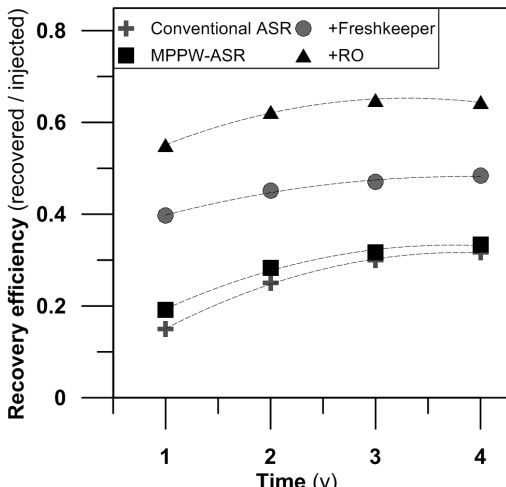

**Figure 12: Recovery efficiencies at the Westland ASR site with and without the borehole leakage resulting from the SEAWAT groundwater transport model for a conventional ASR well (one well screen, fully penetrating), deep injection and shallow recovery via multiple partially penetrating wells without a 'Freshkeeper' (scenario MPPW), for a MPPW in combination with a 'Freshkeeper' (scenario Freshkeeper), and for a scenario in which RO is applied on the intercepted brackish water to produce additional freshwater (50% of the abstracted brackish water).**

The introduction of the Freshkeeper to protect the shallow recovery wells by interception of this deeper saltwater significantly extended the recovery period, enabling recovery of 40% in the first year for direct use. Ultimately, this will yield a RE of almost 50% of virtually unmixed (Cl <50 mg/l) injected freshwater in Cycle 4 in a case without leakage. This will require interception of 18,500 m³ (Cycle 4) to 21,500 m³ of brackish-saline groundwater, such that almost 30,000 m³ of freshwater can be recovered.

When this ASR operational scheme with the Freshkeeper was applied to the field pilot, where short-circuiting saltwater hampered freshwater recovery, approximately one-third of the injected freshwater could be recovered. The ASR-well close to the leaking borehole (AW1) was unable to abstract freshwater in this case. Only AW2 could be used for freshwater recovery, in the end only via the shallowest well (AW2S1). The freshwater loss by short-circuiting cannot be eliminated completely since a large volume of unmixed freshwater is abstracted together with intruding saltwater during the required. The RE will therefore remain lower than in an undisturbed geological setting (RE: 48%). At the same time, the required interception of brackish-saline water will be higher (Table 4), with a total volume of more than 30,000 m³, while around 20,000 m³ of freshwater is recovered.



## 4    Discussion

### 4.1    Saltwater intrusion during the Westland ASR pilot

In this study, the first focus was on the causes for the significantly lower observed freshwater RE of the system. This RE was initially less than a few percent, whereas recovery of around one-third of the injected water was expected. The hydrochemical analyses clearly indicated that the observed salinization was caused by unexpected intrusion of deeper saltwater, as marked by substantially higher $SO_4$ concentrations, which could not be caused by arrival of saltwater from the target aquifer or the upper aquitard, or by the $SO_4$ release upon oxidation of pyrite in the target aquifer. The high $SO_4$-concentrations also exclude early salinization by larger buoyancy effects than initially expected, for instance by a higher K or higher ambient salinities in the target aquifer. The high $SO_4$ concentrations also excluded rapid lateral drift of injected water, as this would also have led to salinization by saltwater with low $SO_4$ concentrations. Additionally, lateral drift would also result in limited REs after addition of the Freshkeeper, which was not the case.

Knowing the source of the salinization, several transport routes can be presumed. First of all, intrusion of deep saltwater may occur when Aquitard 2 has a significantly lower K than derived from grain size analyses, despite the distinct groundwater qualities observed. A more diffuse salinization via Aquitard 2 can then be expected. However, this salinization would be more gradual and better distributed around the wells. It would also mean that Aquitard 2 would quickly freshen during injection and salinize first during recovery. However, the later salinization of Aquitard 2 observed at MW1S4 with respect Aquifer 1 (observed at MW1S3 and K3O1) indicated that Aquitard 2 is by-passed by deeper saltwater during recovery. The presence of (a) conduit(s) therefore provide (a) probable pathway(s) for by-passing saltwater, meaning short-circuiting was occurring between Aquifer 1 and 2. The SEAWAT model underlines that this can indeed explain the early and rapid intrusion by deep saltwater. Since the highest Cl and $SO_4$ concentrations were found in the borehole of the current ATES K3 well (K3O1), this borehole provides the most presumable location of (a) conduit(s). Natural conduits are considered unlikely due to continuity and thickness of Aquitard 2 observed in the surrounding of the ASR wells and the geological genesis (unconsolidated, horizontal lagoonal deposits). The conduit(s) at or around the ATES K3 borehole may originate from the time of installation (improper sealing) or operation, as recorded operation data of the ATES system reports that incidentally exceeded the maximum injection pressure in the well of 1 bar during maintenance in 2009.

### 4.2    The consequences of short-circuiting on ASR in coastal aquifers

The potential effects of short-circuiting induced by deep perturbation on aquifer storage and recovery (ASR) in a shallower coastal aquifer were subsequently explored. In this case of freshwater storage in a confined, saline aquifer, pressure differences induced by the difference in density between injected freshwater and native groundwater provoked intrusion of native groundwater in the injected freshwater bubbles via the presumed conduit. It is illustrated that a complete failure of the ASR system can occur when the short-circuiting via such a conduit occurs close to the ASR wells and little mixing with ambient saltwater is allowed.




The negative effects of short-circuiting on ASR on coastal aquifers are mainly related to the hydraulics around the conduits. First, freshwater is not easily transported downwards through the conduits into a deeper aquifer, while it is easily pushed back into the shallower aquifer when infiltration is stopped or paused. Secondly, the freshwater reaching a deeper aquifer is subjected to buoyancy effects and migrates laterally in the top zone of this deeper aquifer. Finally, during storage and

especially during recovery, the pressure differences in combination with a high hydraulic conductivity rapidly induce a strong flux of saltwater from the whole deeper aquifer into the shallower ASR target aquifer, where a relatively low hydraulic head is present. This short-circuiting induced by such a pressure difference is hampered by the low permeability of the aquitard in a 'pristine situation'. A continuous, undisturbed aquitard is therefore indispensable for the success of ASR in such a setting, as intrusion of deeper saltwater is not desired.

With an increasing distance between the ASR wells and a nearby conduit, the proportion of mixed saltwater in the recovered water decreases while the arrival time increases. When the conduit is situated outside the radius of the injected freshwater body in the target aquifer, a decrease in RE is not expected.

The Westland field example highlights how design, installation, and operational aspects are vital in the more-and-and more exploited subsurface in densely-populated areas. First of all, old boreholes are unreliable and their presence should better be

avoided when selecting new ASR well sites (Maliva et al., 2016). Secondly, installation and operation of (especially injection) wells should be regulated by strict protocols to prevent the creation of new pathways for short-circuiting. Finally, it is important to recognize that similar processes may occur in unperturbed coastal karst aquifers, where natural vertical pipes can be present (Bibby, 1981; Missimer et al., 2002).

### 4.3     Mitigation of short-circuiting on ASR in coastal aquifers

In order to mitigate the short-circuiting and improve the freshwater recovery upon aquifer storage under these unfavourable conditions, several strategies can be recognized. Obviously, sealing of the conduits would be an effective remedy. However, it may not be viable to 1) locate all conduits, for instance when the former wells are decommissioned or when the confining clay layer is fractured upon deeper injection under high pressure, and 2) successfully seal a conduit at a great depth. This underlined by the fact that limited reports of successful sealing of deep conduits can be found.

Apart from sealing, one can also try to deal with these unfavourable conditions. Multiple partially penetrating wells (MPPW) were installed at the Westland ASR site, for instance, enabled interception of intruding saltwater by using the deeper well screens as 'Freshkeepers'. After this intervention, about one-third of virtually unmixed injected freshwater becomes recoverable. This way, the RE is brought to a level similar to the level obtained by an MPPW-equipped ASR system without the Freshkeeper interception and without short-circuiting, while the RE would otherwise remain virtually null. It does require

interception of a significant volume of brackish-saline groundwater, however, which must be injected elsewhere or disposed of.

A significant part of the unmixed freshwater is blended with saltwater in the Freshkeeper wells, such that the freshwater recovery becomes lower than in the situation in which the Freshkeeper is applied and saltwater intrusion via short-circuiting



is absent. At the Westland field site, this is compensated by desalinating the intercepted brackish-saline groundwater, which is a suitable source water for reverse osmosis (RO) thanks to its low salinity. The freshwater (permeate) produced in this process is used for irrigation, while the resulting saltwater (concentrate) is disposed of in Aquifer 2. The resulting RE increase is plotted in Figure 12. Even when no unmixed freshwater is available, desalination of injected water mixed with

groundwater can be continued with this technique to further increase the RE. In comparison with conventional brackish water RO this leads to a better feed water for RO (lower salinity), while salinization of the groundwater system by a net extraction of freshwater is prevented by balancing the freshwater injection and abstraction from the system.

### 4.4     On the performance of ASR in coastal aquifers without leakage: upconing brackish water from the deeper aquitard

In case of a strict water quality limit and relatively saline groundwater, brackish groundwater upconing from the deeper confining aquitard toward shallow recovery wells is a process to take into account, apart from the buoyancy effects in the target aquifer itself. This was shown by the SEAWAT model runs without short-circuiting, which showed a small increase in Cl-concentrations at the ASR wells prior to the full salinization caused by arrival of the fringe of the ASR bubble. The SEAWAT model indicated that the (sandy) clay/peat layer (Aquitard 2) below the target aquifer was the source of upconing

brackish-saline groundwater. Although this layer has a low hydraulic conductivity, it is not impermeable and salinization via diffusion can occur in this zone, while brackish pore water can physically be extracted from this aquitard. The transport processes in this deeper aquitard are comparable with the borehole leakage water via conduits in this aquitard: freshwater is not easily pushed downwards during injection, but brackish water is easily attracted during recovery. After the recovery phase this zone salinizes until the next injection phase starts, so a gradual improvement in time is limited. Brackish water

may also be attracted from the upper aquitard ('downconing'), but this process is counteracted by the buoyancy effects and did not lead to early termination of the freshwater recovery in the Westland case.

The release of brackish water from the deeper aquitard in coastal aquifers can be relevant when quality limits are strict, the native groundwater is saline, and the native groundwater in the target aquifer is displaced far from the ASR wells. The performance of ASR may than be much worse than is predicted by existing ASR performance estimation methods (e.g.

Bakker, 2010; Ward et al., 2009), which assume that impermeable aquitards confine the target aquifer. Even in the first MPPW field test (Zuurbier et al., 2014), this process was not observed, due to a smaller radius of the freshwater bubble, resulting in earlier salinization duo to buoyancy effects. The upconing water can optionally be intercepted by a (small, deep) Freshkeeper well screen to extent the recovery of unmixed freshwater, likewise the interception of intruding saltwater at the Westland site.





## 5    Conclusions

This study shows how short-circuiting negatively affects the freshwater recovery efficiency (RE) during aquifer storage and recovery (ASR) in coastal aquifers. ASR was applied in a shallow saltwater aquifer (23-37 m-BSL) overlying a deeper saltwater aquifer (> 47.5 m-BSL) targeted for aquifer thermal energy storage. Intrusion of deeper saltwater was marked by

chemical tracers (mainly $SO_4$) and quickly terminated the freshwater recovery. The most presumable pathway was the borehole of an ATES well at 3 m from the ASR well (forming a conduit) and was identified by field measurements, hydrochemical analyses, and SEAWAT transport modelling. Transport modelling underlined that the potentially rapid short-circuiting during storage and recovery can reduce the RE to null. This is caused by a rapid intrusion of the deep saltwater already during storage periods, and especially during recovery. Transport modelling also showed that when limited mixing

with ambient groundwater is allowed, a linear RE decrease by short-circuiting with increasing distances from the ASR well within the radius of the injected ASR-bubble is found. Old boreholes should therefore rather be avoided during selection of new ASR sites, or must be situated outside the expected radius.

Field observations and groundwater transport modelling showed that interception of deep short-circuiting water can mitigate the observed RE decrease, although complete compensation of the RE decrease will generally be unattainable since also

unmixed freshwater gets intercepted. At the Westland ASR site, the RE can be brought back to around one-third of the injected water, which is comparable to the RE attained with an ASR system without the Freshkeeper in the same, yet undisturbed setting. With the same Freshkeeper, the set-up would be able to abstract around 50% of the injected water unmixed, if the setting would be undisturbed. This underlines the added value of such a interception well for ASR. Finally, it was found that brackish water upconing from the underlying aquitard towards the shallow recovery wells of the MPPW-ASR

system can occur. In case of strict water quality limits, this process may cause an early termination of freshwater recovery, yet it was neglected in many ASR performance estimations to date.

### Acknowledgements

The authors would like to thank the funding agents of the studies discussed in this paper: Knowledge for Climate, the Joint Water Research Program of the Dutch Water Supply Companies (BTO), and the EU FP7 project 'Demonstrate Ecosystem

Services Enabling Innovation in the Water Sector' (DESSIN, grant agreement no. 619039) and the EU Horizon2020-project 'SUBSOL' (grant agreement no. 642228).

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
