# Peer review of "Consequences and mitigation of saltwater intrusion induced by short-circuiting during aquifer storage and recovery (ASR) in a coastal subsurface"

_Hydrology and Earth System Sciences, 2016_

## Referee Comment (RC1) · D. Pyne (Referee) · 7 Sep 2016

David Pyne Comments on Koen Zuurbier paper September 6, 2016

General: This paper presents an excellent and thorough evaluation of potential mechanisms for salinization of recovered water from ASR storage in brackish aquifers and, in particular, the reduction in recovery efficiency (RE) due to short-circuiting through existing wells open to deeper aquifers. Effective mitigating measures such as the new Freshkeeper technology are of great potential value.

Land use and water quality constraints, and competing uses of coastal aquifer systems
to meet diverse objectives are common to many coastal areas globally. The relatively shallow depths of ASR storage aquifers in the coastal dune sands of the Netherlands are much shallower and more saline than coastal plain alluvial aquifer systems in many other parts of the world. The Dutch ASR experience and associated research therefore extends and deepens the understanding of solutions to similar challenges for ASR wells in other parts of the world. General: It may be helpful to present why the RE was defined for this site by a chloride below 50 mg/l. This makes sense for the Coastal Dunes and greenhouses of the Netherlands, but many readers will wonder why so low compared to drinking water standards, and may tend to be unimpressed by the reported ASR RE values. Page Line 1 26 Key words might also include "upconing" and "downconing," both of which describe "short-circuiting" as referred to in this paper, and as experienced at several operating ASR sites. 2 16 While the impact of short-circuiting may not have been evaluated in the research literature, it has been experienced in the field at many ASR locations in brackish and saline aquifers, resulting in recognition that adequate confinement is usually needed in order to achieve acceptable recovery efficiency (RE). Upconing through underlying confining layers and downconing through overlying confining layers can adversely and rapidly impact RE, whether through an open borehole or through a leaky confining layer.

6 15 For ASR storage of fresh water in a brackish aquifer, with or without the potential for short-circuiting, we would normally provide for initial formation and then mainte-nance of a buffer zone (BZ) to separate the stored fresh water from the surrounding (and underlying or overlying) brackish water. An adequate buffer zone addresses not only blending issues but also geochemical issues such as arsenic attenuation. A typi-cal BZ in an aquifer like this might comprise 30% to 50% of the Target Storage Volume (TSV) that is needed for recovery, however the BZ is a one-time addition of water to the well. Formation of the BZ does not count against recovery efficiency. Instead, it is considered to be a final step in well construction. This typically works….except in situations with short-circuiting where often the BZ volume cannot overcome upcon-ing or downconing of saline water through adjacent open boreholes. Where semi-

permeable confining layers overly or underlie the storage aquifer, we may over time freshen the overlying or underlying adjacent aquifers, thereby steadily improving recovery efficiency.

This paper sets the RE bar very high initially by aiming to recover all of the stored water (60,000 CM). This may be scientifically defensible and convenient however it is effectively counterproductive. Those who provide funds for ASR implementation are more likely to approve projects that achieve higher RE, even if they do not really understand the science. The buffer zone is one of the keys to achieving high RE, and the cost of the water comprising the buffer zone is usually rather small, especially if amortized over the life of the well.

Drawing RE conclusions based on conducting and/or modeling 2 or 3 cycles, without prior formation of a buffer zone, really stacks the deck against achieving satisfactory RE in moderately brackish aquifers. Without short-circuiting, you may be able to achieve a satisfactory RE after maybe 5 to 15 cycles, or with a single cycle following prior development of a buffer zone. The volume of water forming the buffer zone depends on several factors, including leakance of the storage aquifer, lithology and associated dispersion, density differential, aquifer thickness and porosity. We have successfully stored drinking water in a seawater aquifer that had appropriate storage characteristics. 19 4 Consider adding a sentence or two near the beginning of the paper explaining why this site was selected for testing. People would not normally locate an ASR well 3 m from an ATES well that is known to short circuit the lower confining layer. . ..except for the opportunity to conduct research on short circuit mechanisms and associated mitigating measures. 19 25 One bar injection pressure is not very high. We typically limit ASR well injection pressures to about two bars, primarily to avoid wellhead transducer seal failures.

---

## Referee Comment (RC2) · Anonymous Referee #2 · 4 Oct 2016

General comments This manuscript describes solute transport modelling to evaluate the processes that contribute to salinisation of recovered water from an ASR well, along with possible mitigation strategies. Notably, the recovery efficiency of this ASR system is constrained by an extremely tight limit for chloride concentration in the recovered water of 50 mg/L. The content of this manuscript is suitable for publication in HESS, however considerable revision is required to provide the reader with adequate understanding of the site and the evaluation. It should be noted that this ASR system is not typical of ASR operations internationally.

[Figure]

General editing is required to improve the fluency and precision of the language used in the manuscript; particular attention should be paid to improved use of technical descriptions (some examples are outlined in the specific comments below). While this paper stemmed from an unplanned activity (high salinity), the paper should be presented in an organised framework; the current version reads as a somewhat haphazard approach.

Please refer to the specific comments outlined below.

Specific comments

Page Line Comment 1 14 Suggest insert 'confined' into description of aquifers. 1 14 And elsewhere. Suggest replacing 'saltwater' with 'brackish'. 1 16 Explain 'properly separated' – the role of the confining layer should be made clear. 1 17 Suggest replacing SEAWAT with ' variable-density solute transport modelling (SEAWAT)'. 1 22 MPPW has not been defined. 1 28-30 Explain the role of the confining layer and why confined aquifers are targeted for storage. Introduce impact of inter-aquifer leakage. 1 28-30 First sentence needs revision to ensure it is relevant to content of manuscript – stormwater infiltration is also freshwater storage. Perhaps only include uses of confined aquifer for clarity. ASR is one managed aquifer recharge technique suited to storing fresh water in brackish aquifers.

2 1 Over citation of Bonte – revise. 2 Introduction Previous research on inter-aquifer leakage and modelling of recovery efficiency has not been adequately reviewed. 2 5 Of what wells? All wells globally? 2 7 Soil fractures should be described as fractures (aquifer is not soil zone). 2 18 Is it necessary to cite 3 Zuurbier papers? Expand review of literature as many others have stored freshwater in confined aquifer, a necessary reference is R.D.G. Pyne's Aquifer Storage Recovery: A Guide to Groundwater Recharge through Wells. 3 3 Suspect error in number, 270,000 not 270,0000. 3 Method An overview of the hydrogeology is required. Suggesting moving section 3.1 to methods. 3 Method A summary of ASR cycles is required. Suggest adding a table with dates,

injection and recovery volumes for each ASR well. 3 3-13 The reader needs to understand all characteristics of the site, such as aquifer targeted for ASR and for ATES, EC, Cl, SO4 concentrations for each aquifer and for roof runoff. Detail in text is not adequate. 3 8 Need information about hydrogeology before describing the need for MPPW. Screen intervals for all wells should be given in text or table. Distance from ASR wells should be given for ATES and monitoring wells. 3 8 Give distance of ATES wells from ASR well, clearly identify new and old, suggest giving unique identifier (K3-a and K3-b) 3 13 Give the chloride concentrations for each end-member and quantify the amount of mixing <14 1 The monitoring wells shown in Figure 2 have not been mentioned in the text. 4 5 Give sampling frequency. 4 6 Were stable field parameters also used to indicate adequate purging? 4 16 Suggest replacing 'electronic recording' with 'continuous monitoring'. 4 21 Suggest deleting 'In'. 4 21 Give some explanation of model choice. 4 21 Sentence requires editing, move 'to' to after Chiang 2012. 5 1 Replace 'te' with 'the'. 5 8 Justify choice of end-member concentrations; text should be more specific rather than general description. 5 Overall The description of the modelling method could be improved by stating a set of clear aims for the modelling. 6 Table 1 Suggest removing VANI heading. Suggest adding more detail to show properties of each layer. 7 Table 2 Suggest replace 'infiltration' with 'injection'. 7 26 'close by' is vague, be specific. 7 Section 3.1 As mentioned previously, suggest moving to methods. 9 2 As mentioned previously, give summary of ASR cycles in Table. 9 Figure 5 Discussion on mixing is based on Cl, but you are showing EC in Figure. Suggest plotting EC, Cl and SO4. 9 Figure 5 Suspect part a y-axis should be volume not EC. 9 2-8 Give reader more detail, explain that mixing would be expected on fringe of injected bubble and therefore become evident toward end of recovery cycle. 9 2-8 Explain the freshening process in monitoring wells before describing salinisation during recovery. 10 10 Were any other tracers used to identify leakage? Age tracers? 17 Table 4 Recovery of 10018 11 Have the authors considered the economic feasibility of such a scheme? Interception of a significant volume of water must lead to significant pumping costs. What is the driver for such a scheme? 23 16 Bonte et al 2014 not cited References Some editing required,

script errors evident for CO2.

---

## Author Comment (AC1) · 1 Nov 2016

Final author comments on behalf of all co-authors

Referee #1: David Pyne General: This paper presents an excellent and thorough evaluation of potential mechanisms for salinization of recovered water from ASR storage in brackish aquifers and, in particular, the reduction in recovery efficiency (RE) due to short-circuiting through existing wells open to deeper aquifers. Effective mitigating measures such as the new Freshkeeper technology are of great potential value.

Land use and water quality constraints, and competing uses of coastal aquifer systems

to meet diverse objectives are common to many coastal areas globally. The relatively shallow depths of ASR storage aquifers in the coastal dune sands of the Netherlands are much shallower and more saline than coastal plain alluvial aquifer systems in many other parts of the world. The Dutch ASR experience and associated research therefore extends and deepens the understanding of solutions to similar challenges for ASR wells in other parts of the world. General: It may be helpful to present why the RE was defined for this site by a chloride below 50 mg/l. This makes sense for the Coastal Dunes and greenhouses of the Netherlands, but many readers will wonder why so low compared to drinking water standards, and may tend to be unimpressed by the reported ASR RE values. Author's response: The authors agree that these keywords can be added and are willing to do so. We propose to do is in the current Section 2.1

Page Line 1 26 Key words might also include "upconing" and "downconing," both of which describe "short-circuiting" as referred to in this paper, and as experienced at several operating ASR sites. Author's response: The authors agree that these keywords can be added and are willing to do so.

2 16 While the impact of short-circuiting may not have been evaluated in the research literature, it has been experienced in the field at many ASR locations in brackish and saline aquifers, resulting in recognition that adequate confinement is usually needed in order to achieve acceptable recovery efficiency (RE). Upconing through underlying confining layers and downconing through overlying confining layers can adversely and rapidly impact RE, whether through an open borehole or through a leaky confining layer. Author's response: The authors agree with the referee that there is more evaluation of the confinement of target aquifers for ASR. This study, however, focuses on short-circuiting only (for the sake of clarity). The authors are willing to further emphasize this focus in the introduction and mention the earlier work on confinement for ASR, as stated by the reviewer.

6 15 For ASR storage of fresh water in a brackish aquifer, with or without the potential for short-circuiting, we would normally provide for initial formation and then maintenance of a buffer zone (BZ) to separate the stored fresh water from the surrounding (and underlying or overlying) brackish water. An adequate buffer zone addresses not only blending issues but also geochemical issues such as arsenic attenuation. A typical BZ in an aquifer like this might comprise 30% to 50% of the Target Storage Volume (TSV) that is needed for recovery, however the BZ is a one-time addition of water to the well. Formation of the BZ does not count against recovery efficiency. Instead, it is considered to be a final step in well construction. This typically works....except in situations with short-circuiting where often the BZ volume cannot overcome upconing or downconing of saline water through adjacent open boreholes. Where semipermeable confining layers overly or underlie the storage aquifer, we may over time freshen the overlying or underlying adjacent aquifers, thereby steadily improving recovery efficiency. This paper sets the RE bar very high initially by aiming to recover all of the stored water (60,000 CM). This may be scientifically defensible and convenient however it is effectively counterproductive. Those who provide funds for ASR implementation are more likely to approve projects that achieve higher RE, even if they do not really understand the science. The buffer zone is one of the keys to achieving high RE, and the cost of the water comprising the buffer zone is usually rather small, especially if amortized over the life of the well. Drawing RE conclusions based on conducting and/or modeling 2 or 3 cycles, without prior formation of a buffer zone, really stacks the deck against achieving satisfactory RE in moderately brackish aquifers. Without short-circuiting, you may be able to achieve a satisfactory RE after maybe 5 to 15 cycles, or with a single cycle following prior development of a buffer zone. The volume of water forming the buffer zone depends on several factors, including leakance of the storage aquifer, lithology and associated dispersion, density differential, aquifer thickness and porosity. We have successfully stored drinking water in a seawater aquifer that had appropriate storage characteristics. Author's response: The authors recognize the potential additional value of introducing a buffer zone for particular settings, especially when short-circuiting and strong buoyancy effects are absent. Its evaluation was however not the scope of this study, so it cannot be stated what the potential benefit may be

in this case. However, the authors wish to state that given target aquifer characteristics and relatively small scale of the scheme, the (especially long-term) benefits of a buffer zone will be limited due to severe buoyancy effects, which causes the buffer zone to move to the top of the target aquifer, allowing salinization at the aquifer's base. Furthermore, the greenhouse ASR systems do not have any extra water to form a buffer zone: the available rainfall is infiltrated, the water demand is recovered. This study explores this maximum demand in the first cycles, in which the biggest performance increase occurs (Bakker, 2010). For the international perception of ASR, it is very important to note to properly sketch these conditions in order to manage performance expectations of the public. This is a very valid point of the referee. A section on this should be added in the discussion.

19 4 Consider adding a sentence or two near the beginning of the paper explaining why this site was selected for testing. People would not normally locate an ASR well 3 m from an ATES well that is known to short circuit the lower confining layer….except for the opportunity to conduct research on short circuit mechanisms and associated mitigating measures. Author's response: In fact, there was no other location available and the short-circuiting was not known at the time of the realization of the ASR wells. This information should be added to the methods section.

19 25 One bar injection pressure is not very high. We typically limit ASR well injection pressures to about two bars, primarily to avoid wellhead transducer seal failures. Author's response: For relatively shallow (50 m below surface level) injection wells in an area where pressure head are near the land's surface, 1 bar actually already high. Higher injection pressures than 1 bar can already result in failure of the sealing (Olsthoorn, 1981). No amendments suggested.

Please also note the supplement to this comment:
http://www.hydrol-earth-syst-sci-discuss.net/hess-2016-343/hess-2016-343-AC1-supplement.pdf

[Figure]

**Supplement:**

**Final author comments on behalf of all co-authors**

**Referee #1: David Pyne**

General: This paper presents an excellent and thorough evaluation of potential mechanisms for salinization of recovered water from ASR storage in brackish aquifers and, in particular, the reduction in recovery efficiency (RE) due to short-circuiting through existing wells open to deeper aquifers. Effective mitigating measures such as the new Freshkeeper technology are of great potential value.

Land use and water quality constraints, and competing uses of coastal aquifer systems to meet diverse objectives are common to many coastal areas globally. The relatively shallow depths of ASR storage aquifers in the coastal dune sands of the Netherlands are much shallower and more saline than coastal plain alluvial aquifer systems in many other parts of the world. The Dutch ASR experience and associated research therefore extends and deepens the understanding of solutions to similar challenges for ASR wells in other parts of the world.

General: It may be helpful to present why the RE was defined for this site by a chloride below 50 mg/l. This makes sense for the Coastal Dunes and greenhouses of the Netherlands, but many readers will wonder why so low compared to drinking water standards, and may tend to be unimpressed by the reported ASR RE values.

**Author's response:**

*The authors agree that these keywords can be added and are willing to do so. We propose to do is in the current Section 2.1*

Page Line

   1   26      Key words might also include "upconing" and "downconing," both of which describe "short-circuiting" as referred to in this paper, and as experienced at several operating ASR sites.

**Author's response:**

*The authors agree that these keywords can be added and are willing to do so.*

   2   16      While the impact of short-circuiting may not have been evaluated in the research literature, it has been experienced in the field at many ASR locations in brackish and saline aquifers, resulting in recognition that adequate confinement is usually needed in order to achieve acceptable recovery efficiency (RE). Upconing through underlying confining layers and downconing through overlying confining layers can adversely and rapidly impact RE, whether through an open borehole or through a leaky confining layer.

**Author's response:**

*The authors agree with the referee that there is more evaluation of the confinement of target aquifers for ASR. This study, however, focuses on short-circuiting only (for the sake of clarity). The authors are willing to further emphasize this focus in the introduction and mention the earlier work on confinement for ASR, as stated by the reviewer.*

6    15    For ASR storage of fresh water in a brackish aquifer, with or without the potential for short-circuiting, we would normally provide for initial formation and then maintenance of a buffer zone (BZ) to separate the stored fresh water from the surrounding (and underlying or overlying) brackish water. An adequate buffer zone addresses not only blending issues but also geochemical issues such as arsenic attenuation.  A typical BZ in an aquifer like this might comprise 30% to 50% of the Target Storage Volume (TSV) that is needed for recovery, however the BZ is a one-time addition of water to the well.  Formation of the BZ does not count against recovery efficiency.  Instead, it is considered to be a final step in well construction.  This typically works….except in situations with short-circuiting where often the BZ volume cannot overcome upconing or downconing of saline water through adjacent open boreholes.  Where semi-permeable confining layers overly or underlie the storage aquifer, we may over time freshen the overlying or underlying adjacent aquifers, thereby steadily improving recovery efficiency.

This paper sets the RE bar very high initially by aiming to recover all of the stored water (60,000 CM). This may be scientifically defensible and convenient however it is effectively counterproductive. Those who provide funds for ASR implementation are more likely to approve projects that achieve higher RE, even if they do not really understand the science.  The buffer zone is one of the keys to achieving high RE, and the cost of the water comprising the buffer zone is usually rather small, especially if amortized over the life of the well.

Drawing RE conclusions based on conducting and/or modeling 2 or 3 cycles, without prior formation of a buffer zone, really stacks the deck against achieving satisfactory RE in moderately brackish aquifers. Without short-circuiting, you may be able to achieve a satisfactory RE after maybe 5 to 15 cycles, or with a single cycle following prior development of a buffer zone.  The volume of water forming the buffer zone depends on several factors, including leakance of the storage aquifer, lithology and associated dispersion, density differential, aquifer thickness and porosity.  We have successfully stored drinking water in a seawater aquifer that had appropriate storage characteristics.

**Author's response:** *The authors recognize the potential additional value of introducing a buffer zone for particular settings, especially when short-circuiting and strong buoyancy effects are absent. Its evaluation was however not the scope of this study, so it cannot be stated what the potential benefit may be in this case. However, the authors wish to state that given target aquifer characteristics and relatively small scale of the scheme, the (especially long-term) benefits of a buffer zone will be limited due to severe buoyancy effects, which causes the buffer zone to move to the top of the target aquifer, allowing salinization at the aquifer's base. Furthermore, the greenhouse ASR systems do not have any extra water to form a buffer zone: the available rainfall is infiltrated, the water demand is recovered.*

*This study explores this maximum demand in the first cycles, in which the biggest performance increase occurs (Bakker, 2010). For the international perception of ASR, it is very important to note to properly sketch these conditions in order to manage performance expectations of the public. This is a very valid point of the referee. A section on this should be added in the discussion.*

19  4    Consider adding a sentence or two near the beginning of the paper explaining why this site was selected for testing.  People would not normally locate an ASR well 3 m from an ATES well that is known to short circuit the lower confining layer….except for the opportunity to conduct research on short circuit mechanisms and associated mitigating measures.

**Author's response:**

*In fact, there was no other location available and the short-circuiting was not known at the time of the realization of the ASR wells. This information should be added to the methods section.*

19  25  One bar injection pressure is not very high.  We typically limit ASR well injection pressures to about two bars, primarily to avoid wellhead transducer seal failures.

**Author's response:**

*For relatively shallow (50 m below surface level) injection wells in an area where pressure head are near the land's surface, 1 bar actually already high. Higher injection pressures than 1 bar can already result in failure of the sealing (Olsthoorn, 1981). No amendments suggested.*

**Referee #2 (Anonymous)**

General comments

This manuscript describes solute transport modelling to evaluate the processes that contribute to salinisation of recovered water from an ASR well, along with possible mitigation strategies. Notably, the recovery efficiency of this ASR system is constrained by an extremely tight limit for chloride concentration in the recovered water of 50 mg/L. The content of this manuscript is suitable for publication in HESS, however considerable revision is required to provide the reader with adequate understanding of the site and the evaluation. It should be noted that this ASR system is not typical of ASR operations internationally. General editing is required to improve the fluency and precision of the language used in the manuscript; particular attention should be paid to improved use of technical descriptions (some examples are outlined in the specific comments below). While this paper stemmed from an unplanned activity (high salinity), the paper should be presented in an organised framework; the current version reads as a somewhat haphazard approach.

**Author's response:**

*The authors agree that the current version reads somewhat like a 'haphazard' approach. In some way, the study was haphazard: it was never the intention to study the effect of short-circuiting on ASR. It was the intention to increase freshwater recovery upon aquifer storage, despite buoyancy effects, using the multiple partially penetrating wells. However, the authors feel that it is relevant to report failures and surprises during the use of ASR, which is now not always the case (page 2, line 28). To make sure the reader is able to understand and follow the set-up of the system and the operational/monitoring approach, a rather chronological set-up was chosen.*

Please refer to the specific comments outlined below.

Specific comments

Page – Line - Comment

1 14 Suggest insert 'confined' into description of aquifers.

**Author's response:** *will be amended as such*

1 14 And elsewhere. Suggest replacing 'saltwater' with 'brackish'.

**Author's response:** *This is a very common discussions among hydrologists in this field and related to universal definition of freshwater, brackish water, and saltwater. Following the frequently used classification by Stuyfzand (1993), brackish water is limited to water with Cl concentrations <1.000 mg/l. We propose not to amend this choice of word.*

1 16 Explain 'properly separated' – the role of the confining layer should be made clear.

**Author's response:** *We agree with the referee. We propose to add further explanation here.*

1 17 Suggest replacing SEAWAT with ' variable-density solute transport modelling (SEAWAT)'.

**Author's response:** *We agree with the referee and propose to amend this as such*

1 22 MPPW has not been defined.

**Author's response:** *We agree with the referee, MPPW should first be defined before using its abbreviation. This should be done in the same sentence.*

1 28-30 Explain the role of the confining layer and why confined aquifers are targeted for storage. Introduce impact of inter-aquifer leakage.

**Author's response:** *We agree with the referee and propose to add this information in this first part of the introduction to highlight the importance of the confining layer in these applications.*

1 28-30 First sentence needs revision to ensure it is relevant to content of manuscript –stormwater infiltration is also freshwater storage. Perhaps only include uses of confined aquifer for clarity. ASR is one managed aquifer recharge technique suited to storing fresh water in brackish aquifers.

**Author's response:** *It is a bit unclear what the exact point of the referee is here. We propose to replace 'Aquifers' by 'Confined and semi-confined aquifers'. Storm water infiltration is not necessarily storage, but rather 'disposal'. We suggest to combine this with brine disposal.*

2 1 Over citation of Bonte – revise.

**Author's response:** *We found that the most recent and relevant work in this relatively new field was performed by Bonte, but can imagine to mention only the most relevant 2 of its papers.*

2 Introduction Previous research on inter-aquifer leakage and modelling of recovery efficiency has not been adequately reviewed.

**Author's response:** *It is unclear to the authors which other literature is exactly meant here, but the authors are willing to extent the review of inter-aquifer leakage and RE.*

2 5 Of what wells? All wells globally?

**Author's response:** *This concerns all wells worldwide. We suggest to add 'worldwide' after 'wells'*

2 7 Soil fractures should be described as fractures (aquifer is not soil zone).

**Author's response:** *We propose to amend as such*

2 18 Is it necessary to cite 3 Zuurbier papers? Expand review of literature as many others have stored freshwater in confined aquifer, a necessary reference is R.D.G. Pyne's Aquifer Storage Recovery: A Guide to Groundwater Recharge through Wells.

**Author's response:** *We propose to diversify the references here, amongst others by mentioning Pyne (2005) here as well (although it was already in line 31 of page 1 as well).*

3 3 Suspect error in number, 270,000 not 270,0000.

**Author's response:** *Indeed, will be amended as such.*

3 Method An overview of the hydrogeology is required. Suggesting moving section 3.1 to methods.

**Author's response:** *We have had similar considerations, but decided to put this in Section 3 as the geological characterization contains many newly obtained results. However, we agree that this may better suit in section 3.1.*

3 Method A summary of ASR cycles is required. Suggest adding a table with dates, injection and recovery volumes for each ASR well.

**Author's response:** *We agree with the referee. Although the operations during the cycles are reported in Section 3, a summary of the ASR-cycles is welcome in section 2.*

3 3-13 The reader needs to understand all characteristics of the site, such as aquifer targeted for ASR and for ATES, EC, Cl, SO4 concentrations for each aquifer and for roof runoff. Detail in text is not adequate.

**Author's response:** *We feel that we can give the reader the desired detail when we move Section 3.1 to the methods section, with a minor extension.*

3 8 Need information about hydrogeology before describing the need for MPPW. Screen intervals for all wells should be given in text or table. Distance from ASR wells should be given for ATES and monitoring wells. 3 8 Give distance of ATES wells from ASR well, clearly identify new and old, suggest giving unique identifier (K3-a and K3-b)

**Author's response:** *The authors agree with the referee and are willing to provide this information*

3 13 Give the chloride concentrations for each end-member and quantify the amount of mixing <

**Author's response:** *We agree with the reviewer and propose to add this information*

14 1 The monitoring wells shown in Figure 2 have not been mentioned in the text.

**Author's response:** *We propose to add this information in the table requested by the reviewer*

4 5 Give sampling frequency.

**Author's response:** *We propose to add the exact moments of sampling*

4 6 Were stable field parameters also used to indicate adequate purging?

**Author's response:** *Yes, upon purging of each well, it was checked if all field parameters were stable. We will add this to information to this section.*

4 16 Suggest replacing 'electronic recording' with 'continuous monitoring'.

**Author's response:** *We can see an improved clarity by implementing this suggestion and are willing to do so.*

4 21 Suggest deleting 'In'. 4 21 Give some explanation of model choice.

**Author's response:** *We agree with the referee, this was a typo. We will also elaborate on the choice for SEAWAT (variable density, etc.)*

4 21 Sentence requires editing, move 'to' to after Chiang 2012.

**Author's response:** *Indeed, a second typo in this sentence. Amended as such.*

5 1 Replace 'te' with 'the'.

**Author's response:** *Typo, amended as such.*

5 8 Justify choice of end-member concentrations; text should be more specific rather than general description.

**Author's response:** *We propose to add justification here, as proposed by the referee.*

5 Overall The description of the modelling method could be improved by stating a set of clear aims for the modelling.

**Author's response:** *The authors agree that the reader should be taken by the hand in this section by elaborating on the modelling aims and propose to add the following:*

*'Groundwater transport modelling was executed to validate the added value of the MPPW set-up under the local conditions. In the later stage of the research, the groundwater transport model was used to test potential pathways for deeper groundwater to enter the target aquifer and explore the characteristics of a potential conduit via scenario modelling. Correction for groundwater densities in the flow modelling was vital, due to significant contrast between the aquifer's saltwater and the injected rainwater.'*

6 Table 1 Suggest removing VANI heading. Suggest adding more detail to show properties of each layer.

**Author's response:** *The authors feel that the table (including VANI) provides the relevant characteristics of the model layers. VANI can be replaced by reporting the $K_v$ in this column*

7 Table 2 Suggest replace 'infiltration' with 'injection'.

**Author's response:** *For the sake of clarity, we tend to use 'infiltration' instead of 'injection' since the system does not directly pump the water into the aquifer, but to a 4m high standpipe. This standpipe is connected to the ASR wells. The water therefore 'infiltrates' (rather slowly) in to the aquifer. The reader may have a more 'violent' perspective of infiltration*

7 26 'close by' is vague, be specific.

**Author's response:** *The authors agree and propose to replace 'close by pumping test' by 'a pumping test at approximately 500 m from the ASR wells,'*

7 Section 3.1 As mentioned previously, suggest moving to methods.

**Author's response:** *The authors agree, as indicated.*

9 2 As mentioned previously, give summary of ASR cycles in Table.

**Author's response:** *The authors agree, as indicated.*

9 Figure 5 Discussion on mixing is based on Cl, but you are showing EC in Figure. Suggest plotting EC, Cl and SO4.

**Author's response:** *We thank the reviewer for this suggestion. However, the reason to plot the EC here is the extremely high frequency of the analyses, which give far more information that the regularly measured*

9 Figure 5 Suspect part a y-axis should be volume not EC.

**Author's response:** *Indeed, the authors will amend the title on this axis (correct = pumped volume ($m^3$)*

9 2-8 Give reader more detail, explain that mixing would be expected on fringe of injected bubble and therefore become evident toward end of recovery cycle.

**Author's response:** *The author realize that in this section somewhat more information on what was expected, such that the reader can better understand the deviations and are willing to add this information.*

9 2-8 Explain the freshening process in monitoring wells before describing salinisation during recovery.

**Author's response:** *The authors agree, this also relates to the previous comment.*

10 10 Were any other tracers used to identify leakage? Age tracers?

**Author's response:** *As indicated in the Methods Section, only macrochemical analyses were performed. Especially $SO_4$ showed to be a good tracer, followed by HCO3 (which however had a much lower contrast. Na amendments suggested.*

17 Table 4 Recovery of 10018

**Author's response:** *It is unclear to the authors what is meant by the reviewer here.*

11 Have the authors considered the economic feasibility of such a scheme? Interception of a significant volume of water must lead to significant pumping costs. What is the driver for such a scheme?

**Author's response:** *Up to now, an extensive evaluation of the costs of this scheme has not been executed. However, it is known that due to low price of electricity in The Netherlands, this impact will be limited.*

23 16 Bonte et al 2014 not cited

**Author's response:** *Citation removed. This reference incidentally ended up in the reference section.*

References Some editing required, script errors evident for CO2.

**Author's response:** *This was due to some problems with Endnote and the desired format of HESS. Will be corrected.*

---

## Author Response (AR1)

**Final author comments on behalf of all co-authors**

**Referee #1: David Pyne**

General: This paper presents an excellent and thorough evaluation of potential mechanisms for salinization of recovered water from ASR storage in brackish aquifers and, in particular, the reduction in recovery efficiency (RE) due to short-circuiting through existing wells open to deeper aquifers. Effective mitigating measures such as the new Freshkeeper technology are of great potential value.

Land use and water quality constraints, and competing uses of coastal aquifer systems to meet diverse objectives are common to many coastal areas globally. The relatively shallow depths of ASR storage aquifers in the coastal dune sands of the Netherlands are much shallower and more saline than coastal plain alluvial aquifer systems in many other parts of the world. The Dutch ASR experience and associated research therefore extends and deepens the understanding of solutions to similar challenges for ASR wells in other parts of the world.

General: It may be helpful to present why the RE was defined for this site by a chloride below 50 mg/l. This makes sense for the Coastal Dunes and greenhouses of the Netherlands, but many readers will wonder why so low compared to drinking water standards, and may tend to be unimpressed by the reported ASR RE values.

**Author's response:**

*The authors agree that these keywords can be added and are willing to do so. We propose to do is in the current Section 2.1*

Page  Line

1 26 Key words might also include "upconing" and "downconing," both of which describe "short-circuiting" as referred to in this paper, and as experienced at several operating ASR sites.

**Author's response:**

*The authors agree and have added these keywords*

2 16 While the impact of short-circuiting may not have been evaluated in the research literature, it has been experienced in the field at many ASR locations in brackish and saline aquifers, resulting in recognition that adequate confinement is usually needed in order to achieve acceptable recovery efficiency (RE). Upconing through underlying confining layers and downconing through overlying confining layers can adversely and rapidly impact RE, whether through an open borehole or through a leaky confining layer.

**Author's response:**

*The authors agree with the referee that there is more evaluation of the confinement of target aquifers for ASR. This study, however, focuses on short-circuiting only (for the sake of clarity). The authors feel that this focus is clear in the introduction. The authors now mention the earlier work on confinement for ASR, as stated by the reviewer.*

6    15    For ASR storage of fresh water in a brackish aquifer, with or without the potential for short-circuiting, we would normally provide for initial formation and then maintenance of a buffer zone (BZ) to separate the stored fresh water from the surrounding (and underlying or overlying) brackish water. An adequate buffer zone addresses not only blending issues but also geochemical issues such as arsenic attenuation.  A typical BZ in an aquifer like this might comprise 30% to 50% of the Target Storage Volume (TSV) that is needed for recovery, however the BZ is a one-time addition of water to the well.  Formation of the BZ does not count against recovery efficiency.  Instead, it is considered to be a final step in well construction.  This typically works….except in situations with short-circuiting where often the BZ volume cannot overcome upconing or downconing of saline water through adjacent open boreholes.  Where semi-permeable confining layers overly or underlie the storage aquifer, we may over time freshen the overlying or underlying adjacent aquifers, thereby steadily improving recovery efficiency.

This paper sets the RE bar very high initially by aiming to recover all of the stored water (60,000 CM). This may be scientifically defensible and convenient however it is effectively counterproductive. Those who provide funds for ASR implementation are more likely to approve projects that achieve higher RE, even if they do not really understand the science.  The buffer zone is one of the keys to achieving high RE, and the cost of the water comprising the buffer zone is usually rather small, especially if amortized over the life of the well.

Drawing RE conclusions based on conducting and/or modeling 2 or 3 cycles, without prior formation of a buffer zone, really stacks the deck against achieving satisfactory RE in moderately brackish aquifers. Without short-circuiting, you may be able to achieve a satisfactory RE after maybe 5 to 15 cycles, or with a single cycle following prior development of a buffer zone.  The volume of water forming the buffer zone depends on several factors, including leakance of the storage aquifer, lithology and associated dispersion, density differential, aquifer thickness and porosity.  We have successfully stored drinking water in a seawater aquifer that had appropriate storage characteristics.

**Author's response:** *The authors recognize the potential additional value of introducing a buffer zone for particular settings, especially when short-circuiting and strong buoyancy effects are absent. Its evaluation was however not the scope of this study, so it cannot be stated what the potential benefit may be in this case. However, the authors wish to state that given target aquifer characteristics and relatively small scale of the scheme, the (especially long-term) benefits of a buffer zone will be limited due to severe buoyancy effects, which causes the buffer zone to move to the top of the target aquifer, allowing salinization at the aquifer's base. Furthermore, the greenhouse ASR systems do not have any extra water to form a buffer zone: the available rainfall is infiltrated, the water demand is recovered.*

*This study explores this maximum demand in the first cycles, in which the biggest performance increase occurs (Bakker, 2010). For the international perception of ASR, it is very important to note to properly sketch these conditions in order to manage performance expectations of the public. This is a very valid point of the referee. A section on this should be added in the discussion (Section 4.4):*

> *'Finally it should be noted that the ASR-system analyzed in this study had very strict water quality limits (practically no mixing allowed) and that a buffer zone (Pyne, 2005) between the injected freshwater and the relatively saline ambient groundwater was not realized before starting the ASR-cycles. The boundary conditions for ASR were therefore already unfavorable. Also, the potential improvement after >3 cycles was not explored. The performance of this ASR-system should therefore not be considered the typical performance of ASR in a brackish-saline aquifers, which controlled by a complex interplay of geological conditions and operational*

*parameters (Bakker, 2010), well design (Zuurbier et al., 2015; Zuurbier et al., 2014), and the formation of a buffer zone prior to starting the ASR-cycles (Pyne, 2005).'*

19  4    Consider adding a sentence or two near the beginning of the paper explaining why this site was selected for testing. People would not normally locate an ASR well 3 m from an ATES well that is known to short circuit the lower confining layer….except for the opportunity to conduct research on short circuit mechanisms and associated mitigating measures.

**Author's response:**

*In fact, there was no other location available and the short-circuiting was not known at the time of the realization of the ASR wells. This information was added to the methods section (Section 2.1).*

19  25   One bar injection pressure is not very high. We typically limit ASR well injection pressures to about two bars, primarily to avoid wellhead transducer seal failures.

**Author's response:**

*For relatively shallow (50 m below surface level) injection wells in an area where pressure head are near the land's surface, 1 bar actually already high. Higher injection pressures than 1 bar can already result in failure of the sealing (Olsthoorn, 1981). No amendments suggested.*

**Referee #2 (Anonymous)**

General comments

This manuscript describes solute transport modelling to evaluate the processes that contribute to salinisation of recovered water from an ASR well, along with possible mitigation strategies. Notably, the recovery efficiency of this ASR system is constrained by an extremely tight limit for chloride concentration in the recovered water of 50 mg/L. The content of this manuscript is suitable for publication in HESS, however considerable revision is required to provide the reader with adequate understanding of the site and the evaluation. It should be noted that this ASR system is not typical of ASR operations internationally. General editing is required to improve the fluency and precision of the language used in the manuscript; particular attention should be paid to improved use of technical descriptions (some examples are outlined in the specific comments below). While this paper stemmed from an unplanned activity (high salinity), the paper should be presented in an organised framework; the current version reads as a somewhat haphazard approach.

**Author's response:**

*The authors agree that the current version reads somewhat like a 'haphazard' approach. In some way, the study was haphazard: it was never the intention to study the effect of short-circuiting on ASR. It was the intention to increase freshwater recovery upon aquifer storage, despite buoyancy effects, using the multiple partially penetrating wells. However, the authors feel that it is relevant to report failures and surprises during the use of*

*ASR, which is now not always the case (page 2, line 28, original manuscript). To make sure the reader is able to understand and follow the set-up of the system and the operational/monitoring approach, a rather chronological set-up was chosen.*

Please refer to the specific comments outlined below.

Specific comments

Page – Line - Comment

1 14 Suggest insert 'confined' into description of aquifers.

**Author's response:** *amended as such*

1 14 And elsewhere. Suggest replacing 'saltwater' with 'brackish'.

**Author's response:** *This is a very common discussions among hydrologists in this field and related to universal definition of freshwater, brackish water, and saltwater. Following the frequently used classification by Stuyfzand (1993), brackish water is limited to water with Cl concentrations <1.000 mg/l. We propose not to amend this choice of word.*

1 16 Explain 'properly separated' – the role of the confining layer should be made clear.

**Author's response:** *We agree with the referee. We added further explanation here by adding after 'separated':*

*'(i.e. a continuous clay layer prevented rapid groundwater flow between both aquifers)'*

1 17 Suggest replacing SEAWAT with ' variable-density solute transport modelling (SEAWAT)'.

**Author's response:** *We agree with the referee and this was amended as such.*

1 22 MPPW has not been defined.

**Author's response:** *We agree with the referee, MPPW should first be defined before using its abbreviation. This is done in the same sentence.*

1 28-30 Explain the role of the confining layer and why confined aquifers are targeted for storage. Introduce impact of inter-aquifer leakage.

**Author's response:** *We agree with the referee and propose to add this information in this first part of the introduction to highlight the importance of the confining layer in these applications. It was added near the references on this topic, which were suggested by Referee #1.*

1 28-30 First sentence needs revision to ensure it is relevant to content of manuscript –stormwater infiltration is also freshwater storage. Perhaps only include uses of confined aquifer for clarity. ASR is one managed aquifer recharge technique suited to storing fresh water in brackish aquifers.

**Author's response:** *It is a bit unclear what the exact point of the referee is here. We propose to replace 'Aquifers' by 'Confined and semi-confined aquifers'. Storm water infiltration is not necessarily storage, but rather 'disposal'. We combined this with brine disposal.*

2 1 Over citation of Bonte – revise.

 **Author's response:** *We found that the most recent and relevant work in this relatively new field was performed by Bonte, but we now mention only the most relevant 2 of its papers.*

2 Introduction Previous research on inter-aquifer leakage and modelling of recovery efficiency has not been adequately reviewed.

**Author's response:** *It is unclear to the authors which other literature is exactly meant here, but the authors are willing to extent the review of inter-aquifer leakage and RE. See also the response to Referee #1 and the added references in the introduction on this matter.*

2 5 Of what wells? All wells globally?

**Author's response:** *This concerns all wells worldwide. We added 'worldwide' after 'wells'*

2 7 Soil fractures should be described as fractures (aquifer is not soil zone).

**Author's response:** *Amended as suggested.*

2 18 Is it necessary to cite 3 Zuurbier papers? Expand review of literature as many others have stored freshwater in confined aquifer, a necessary reference is R.D.G. Pyne's Aquifer Storage Recovery: A Guide to Groundwater Recharge through Wells.

**Author's response:** *We diversified the references here, amongst others by mentioning Pyne (2005) here as well (although it was already in line 31 of page 1 as well).*

3 3 Suspect error in number, 270,000 not 270,0000.

**Author's response:** *Indeed, amended.*

3 Method An overview of the hydrogeology is required. Suggesting moving section 3.1 to methods.

**Author's response:** *We have had similar considerations, but decided to put this in Section 3 as the geological characterization contains many newly obtained results. However, we agree that this may better suit in section 2. Section 3.1 was moved to Section 2.*

3 Method A summary of ASR cycles is required. Suggest adding a table with dates, injection and recovery volumes for each ASR well.

**Author's response:** *We agree with the referee. Although the operations during the cycles are reported in Section 3, a summary of the ASR-cycles presented in a table is now implemented in Section 2.*

3 3-13 The reader needs to understand all characteristics of the site, such as aquifer targeted for ASR and for ATES, EC, Cl, SO4 concentrations for each aquifer and for roof runoff. Detail in text is not adequate.

**Author's response:** *We feel that we give the reader the desired detail by moving Section 3.1 to the methods section.*

3 8 Need information about hydrogeology before describing the need for MPPW. Screen intervals for all wells should be given in text or table. Distance from ASR wells should be given for ATES and monitoring wells.

**Author's response:** *The authors agree with the referee and now provide this information*

3 8 Give distance of ATES wells from ASR well, clearly identify new and old, suggest giving unique identifier (K3-a and K3-b)

**Author's response:** *The distance is now mentioned in Section 2.1. A unique identifier is added as proposed, also in the related figures and the rest of the text.*

3 13 Give the chloride concentrations for each end-member and quantify the amount of mixing <

**Author's response:** *We agree with the reviewer and added this information in Section 2.2.*

14 1 The monitoring wells shown in Figure 2 have not been mentioned in the text.

**Author's response:** *This information is now reported in text and in the table as requested by the reviewer*

4 5 Give sampling frequency.

**Author's response:** *We added the exact moments of sampling*

4 6 Were stable field parameters also used to indicate adequate purging?

**Author's response:** *Yes, upon purging of each well, it was checked if all field parameters were stable. We added this information to this section.*

4 16 Suggest replacing 'electronic recording' with 'continuous monitoring'.

**Author's response:** *We can see an improved clarity by implementing this suggestion. Amended as suggested.*

4 21 Suggest deleting 'In'. 4 21 Give some explanation of model choice.

**Author's response:** *We agree with the referee, this was a typo. We now also elaborate on the choice for SEAWAT (variable density, etc.)*

4 21 Sentence requires editing, move 'to' to after Chiang 2012.

**Author's response:** *Indeed, a second typo in this sentence. Amended as such.*

5 1 Replace 'te' with 'the'.

**Author's response:** *Typo, amended as suggested.*

5 8 Justify choice of end-member concentrations; text should be more specific rather than general description.

**Author's response:** *We added specific justification here, as proposed by the referee.*

5 Overall The description of the modelling method could be improved by stating a set of clear aims for the modelling.

**Author's response:** *The authors agree that the reader should be taken by the hand in this section by elaborating on the modelling aims and added the following at the start of the section:*

*'Groundwater transport modelling was executed to validate the added value of the MPPW set-up under the local conditions. In the later stage of the research, the groundwater transport model was used to test potential pathways for deeper groundwater to enter the target aquifer and explore the characteristics of a potential conduit via scenario modelling. Correction for groundwater densities in the flow modelling was vital, due to significant contrast between the aquifer's saltwater and the injected rainwater.'*

6 Table 1 Suggest removing VANI heading. Suggest adding more detail to show properties of each layer.

**Author's response:** *The authors feel that the table (including VANI) provides the relevant characteristics of the model layers. VANI is replaced by reporting the $K_v$ in this column for the sake of clarity.*

7 Table 2 Suggest replace 'infiltration' with 'injection'.

**Author's response:** *The authors agree with the reviewer and replaced infiltration by injection, throughout the manuscript.*

7 26 'close by' is vague, be specific.

**Author's response:** *The authors agree and propose to replace 'close by pumping test' by 'a pumping test at approximately 500 m from the ASR wells,'*

7 Section 3.1 As mentioned previously, suggest moving to methods.

**Author's response:** *The authors agree, amended as indicated.*

9 2 As mentioned previously, give summary of ASR cycles in Table.

**Author's response:** *The authors agree, amended as indicated.*

9 Figure 5 Discussion on mixing is based on Cl, but you are showing EC in Figure. Suggest plotting EC, Cl and SO4.

**Author's response:** *We thank the reviewer for this suggestion. However, the reason to plot the EC here is the extremely high frequency of the analyses, which gives far more information that the regularly measured Cl.*

9 Figure 5 Suspect part a y-axis should be volume not EC.

**Author's response:** *Indeed, the authors amended the title on this axis (correct = pumped volume ($m^3$/yr)*

9 2-8 Give reader more detail, explain that mixing would be expected on fringe of injected bubble and therefore become evident toward end of recovery cycle.

**Author's response:** *The authors realize that in this section somewhat more information on what was expected, such that the reader can better understand the deviations and added this information in Section 3.1.*

9 2-8 Explain the freshening process in monitoring wells before describing salinisation during recovery.

**Author's response:** *The authors agree, this also relates to the previous comment.*

10 10 Were any other tracers used to identify leakage? Age tracers?

**Author's response:** *As indicated in the Methods Section, only macrochemical analyses were performed. Especially SO$_4$ showed to be a good tracer, followed by HCO3 (which however had a much lower contrast. No amendments suggested.*

17 Table 4 Recovery of 10018

**Author's response:** *It is unclear to the authors what is meant by the reviewer here.*

11 Have the authors considered the economic feasibility of such a scheme? Interception of a significant volume of water must lead to significant pumping costs. What is the driver for such a scheme?

**Author's response:** *Up to now, an extensive evaluation of the costs of this scheme has not been executed. However, it is known that due to low price of electricity in The Netherlands, this impact will be limited. At the end of Section 4.3, some reflection on this matter is added.*

23 16 Bonte et al 2014 not cited

**Author's response:** *Citation removed. This reference incidentally ended up in the reference section.*

References Some editing required, script errors evident for CO2.

**Author's response:** *This was due to some problems with Endnote and the desired format of HESS. Will be corrected during typesetting.*

[revised manuscript text omitted]